# Simultaneous measurement of chromatin accessibility, DNA methylation, and nucleosome phasing in single cells

Sebastian Pott*

Department of Human Genetics, University of Chicago, Chicago, United States

**Abstract** Gaining insights into the regulatory mechanisms that underlie the transcriptional variation observed between individual cells necessitates the development of methods that measure chromatin organization in single cells. Here I adapted *Nucleosome Occupancy and Methylome-sequencing* (NOMe-seq) to measure chromatin accessibility and endogenous DNA methylation in single cells (scNOMe-seq). scNOMe-seq recovered characteristic accessibility and DNA methylation patterns at DNase hypersensitive sites (DHSs). An advantage of scNOMe-seq is that sequencing reads are sampled independently of the accessibility measurement. scNOMe-seq therefore controlled for fragment loss, which enabled direct estimation of the fraction of accessible DHSs within individual cells. In addition, scNOMe-seq provided high resolution of chromatin accessibility within individual loci which was exploited to detect footprints of CTCF binding events and to estimate the average nucleosome phasing distances in single cells. scNOMe-seq is therefore well-suited to characterize the chromatin organization of single cells in heterogeneous cellular mixtures.

## Introduction

Extensive transcriptional variation between individual cells has been observed using single cell RNA-seq. These data facilitate identification of functional subpopulations in seemingly homogeneous cell populations (*Shalek et al., 2014*), or characterization of the cellular composition of complex tissues (*Jaitin et al., 2014*; *Treutlein et al., 2014*; *Macosko et al., 2015*). To gain mechanistic insights into regulatory features that underlie cellular heterogeneity it is essential to measure chromatin organization in individual cells. A number of methods that map chromatin organization in populations of cells have been adapted for single cells, including ATAC-seq (*Cusanovich et al., 2015*; *Buenrostro et al., 2015b*), DNase-seq (*Jin et al., 2015*), methylome sequencing (*Smallwood et al., 2014*; *Farlik et al., 2015*), and ChIP-seq (*Rotem et al., 2015*). Interpretation of these data in single cells is complicated because of the near binary and extremely sparse signal (*Cusanovich et al., 2015*; *Buenrostro et al., 2015b*; *Maurano and Stamatoyannopoulos, 2015*). *Nucleosome Occupancy and Methylome-sequencing* (NOMe-seq) (*Kelly et al., 2012*) employs the GpC methyltransferase (MTase) from *M. CviPI* to probe chromatin accessibility (*Kelly et al., 2012*; *Kilgore et al., 2007*). The GpC MTase methylates cytosines in GpC dinucleotides in non-nucleosomal DNA in vitro. Combined with high-throughput bisulfite sequencing this approach has been used to characterize nucleosome positioning and endogenous methylation in human cell lines (*Kelly et al., 2012*; *Taberlay et al., 2014*) and in selected promoters of single yeast cells (*Small et al., 2014*). NOMe-seq data have several unique features that are advantageous considering the challenges associated with single cell measurements (*Figure 1a*). First, NOMe-seq simultaneously measures chromatin accessibility (through GpC methylation) and endogenous CpG methylation. Chromatin accessibility indicates whether a putative regulatory region might be utilized in a given cell (*ENCODE Project Consortium, 2012*), while endogenous DNA methylation in regulatory regions has been connected to a variety of regulatory

*For correspondence: spott@uchicago.edu

**Competing interests:** The author declares that no competing interests exist.

**eLife digest** DNA contains all the information and instructions needed to build an organism and enables its cells to fulfil their role. There are many different cell types in animals, and each type only needs a small portion of the information found in the DNA to do its job. Hence, only the sections – also known as genes – specific to a particular role need to be active or 'expressed' in any given cell type. The sets of genes that are active in a cell determine what that cell can do and make it different from other cell types. Short regions that surround the genes act as 'switches' to control their activity.

To study how these switches work, researchers have developed techniques that can measure when specific ones are on or off in different cell types. A technique called NOMe-seq can simultaneously identify active and inactive regions in the genome by measuring markers that are specific for active and inactive regions. However, this approach was created for large samples containing thousands of cells, so-called bulk samples. Since the pattern of gene expression can vary between individual cells even if they are of the same cell type, it is important to analyse each individual cell.

Sebastian Pott has now adapted the NOMe-seq technique for use in single cells. The modified protocol was used to study human cells that have been well studied and can be grown in the laboratory. The results of this proof-of-principle study showed that the adapted version of the technique obtained similar results as when used on bulk samples. This demonstrates that the method can reliably measure active and inactive regions in single cells across the genome.

The next step will be to use this tool to study how genes are affected in diseases like cancer, where gene expression can be variable between individual cells. For example, understanding the differences between individual cells within a cancer sample might help to understand why some cancers react to treatments better than others.

processes often associated with repression (*Schübeler, 2015*). The ability to combine complementary assays within single cells is essential for a comprehensive genomic characterization of individual cells since each cell represents a unique biological sample which is almost inevitably destroyed in the process of the measurement. Second, each sequenced read might contain several GpCs which independently report the accessibility status along the length of that read. NOMe-seq therefore captures additional information compared to purely count-based methods, such as ATAC-seq and DNase-seq, which increases the confidence associated with the measurements and allows detection of footprints of individual transcription factor (TF) binding events in single cells. Third, the DNA is recovered and sequenced independently of its methylation status, which is a pre-requisite to distinguish between true negatives (i.e. closed chromatin) and false negatives (i.e. loss of DNA) when assessing accessibility at specified locations in single cells. This is especially important in single cells where allelic drop-out is pervasive. In single cells, NOMe-seq can therefore measure the fraction of accessible regions among a set of covered, pre-defined genomic locations. In this proof-of-principle study, I showed that NOMe-seq, which previously had only been performed on bulk samples (*Kelly et al., 2012*; *Taberlay et al., 2014*), can be performed on single cells. In addition to endogenous methylation at CpG dinucleotides, single cell NOMe-seq (scNOMe-seq) measured chromatin accessibility at DHSs and TF binding sites in individual cells, and detected footprints of CTCF binding at individual loci. Finally, the average phasing distance between nucleosomes within individual cells can also be estimated from scNOMe-seq data.

## Results

To adapt the NOMe-seq protocol (*Kelly et al., 2012*; *Miranda et al., 2010*) to single cells, individual nuclei were first incubated with GpC MTase and then sorted into wells of a 96-well plate using fluorescence-activated cell sorting (FACS) (*Figure 1b* and *Figure 1—figure supplement 1*). DNA from isolated nuclei was subjected to bisulfite conversion and sequencing libraries were prepared using a commercial kit for amplification of low amounts of bisulfite-converted DNA (Materials and methods). To assess the feasibility and performance of NOMe-seq in single cells, I used the well-characterized cell lines GM12878 and K562. The scNOMe-seq datasets in this study represent 19 individual

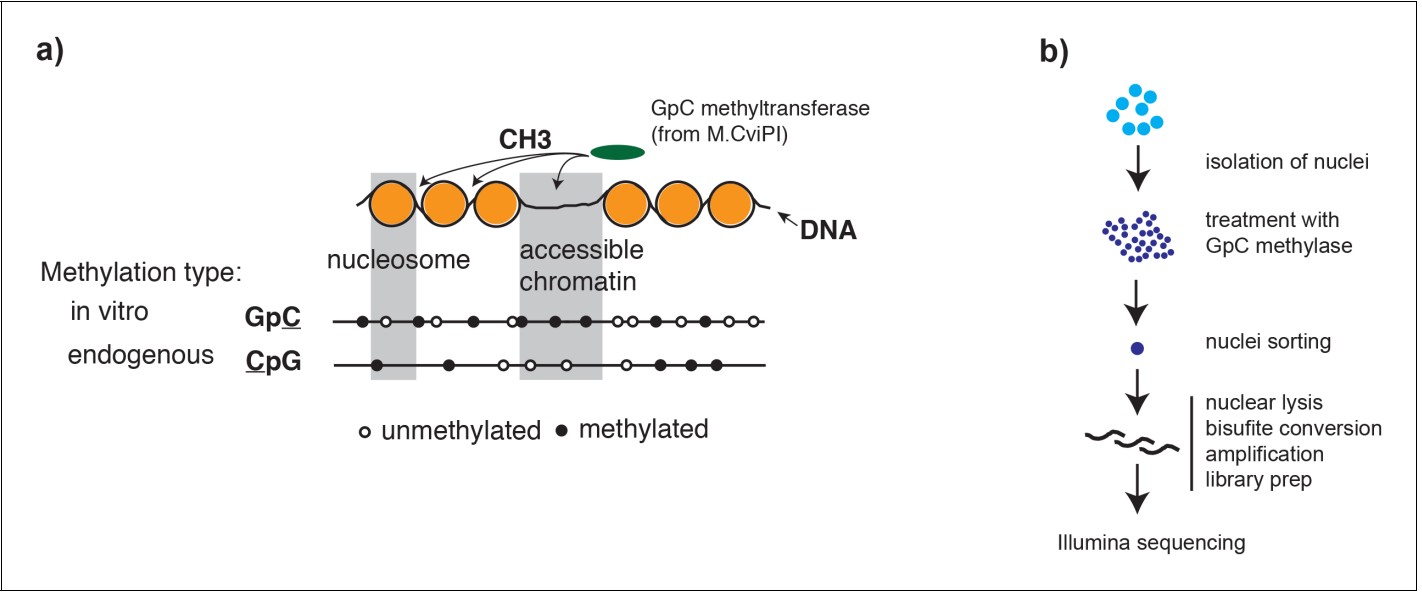

**Figure 1.** Overview of scNOMe-seq procedure. (**a**) Schematic of GpC methyltransferase-based mapping of chromatin accessibility and simultaneous detection of endogenous DNA methylation. (**b**) Schematic of scNOMe-seq procedure introduced in this study.

The following figure supplements are available for figure 1:

**Figure supplement 1.** FACS profile from Hoechst stained nuclei to assess DNA content.

**Figure supplement 2.** Schematic of experimental set up.

**Figure supplement 3.** Number of covered GpC and CpG dinucleotides is proportional to the number of total bases covered.

GM12878 cells and 11 individual K562 cells. The set of GM12878 cells included seven control cells that were not treated with GpC MTase (*Figure 1—figure supplement 2*). Each GpC MTase-treated library was sequenced to at least 16 M individual reads (Materials and methods). Reads were aligned to the human genome using the aligner Bismark (*Krueger et al., 2012*) and, after removal of duplicate reads, between 2.5M and 5M reads were retained per library (*Supplementary file 1*). On average 6,679,864 (2.9%) of all cytosines in GpCs and 1,291,180 (3.6%) of all cytosines in CpGs were covered per cell (*Figure 2—figure supplement 1* and *Supplementary file 1*).

## scNOMe-seq accurately detected accessible chromatin at DNaseI hypersensitive sites

To test whether the GpC methylation observed in GpC MTase treated samples (*Figure 2—figure supplement 1*) captured known chromatin accessibility patterns, I focused on DNaseI hypersensitive sites (DHSs) that were previously identified in GM12878 and K562 cell lines (*ENCODE Project Consortium, 2012*). DHSs were associated with strong enrichment of GpC methylation, both in data from pooled and individual GM12878 (*Figure 2a,b*, *Figure 2—figure supplement 2*) and K562 cells (*Figure 2—figure supplements 3* and *4*). Conversely, endogenous CpG methylation decreased around the center of the DHSs in agreement with previous reports (*Stadler et al., 2011*; *Ziller et al., 2015*) (*Figure 2a* and *Figure 2—figure supplement 3*). These data show that scNOMe-seq detected chromatin accessibility at DHSs. To assess how many of the known DHSs regions were recovered in a single cell, I first filtered DHSs that contained GpC dinucleotides within their primary sequence and thus could be theoretically detected by NOMe-seq. The frequent occurrence of GpC di-nucleotides renders the majority (>85%) of DHSs detectable by NOMe-seq (*Figure 2—figure supplements 5* and *6*). Of the theoretically detectable DHSs, 10.6% (20388/191566) and 17.3% (33182/191598) had one or more GpCs covered and, using a more stringent criterion, 5.2% (9083/174896) and 9.5% (16608/174828) were covered at four or more GpCs in individual GM12878 cells and K562

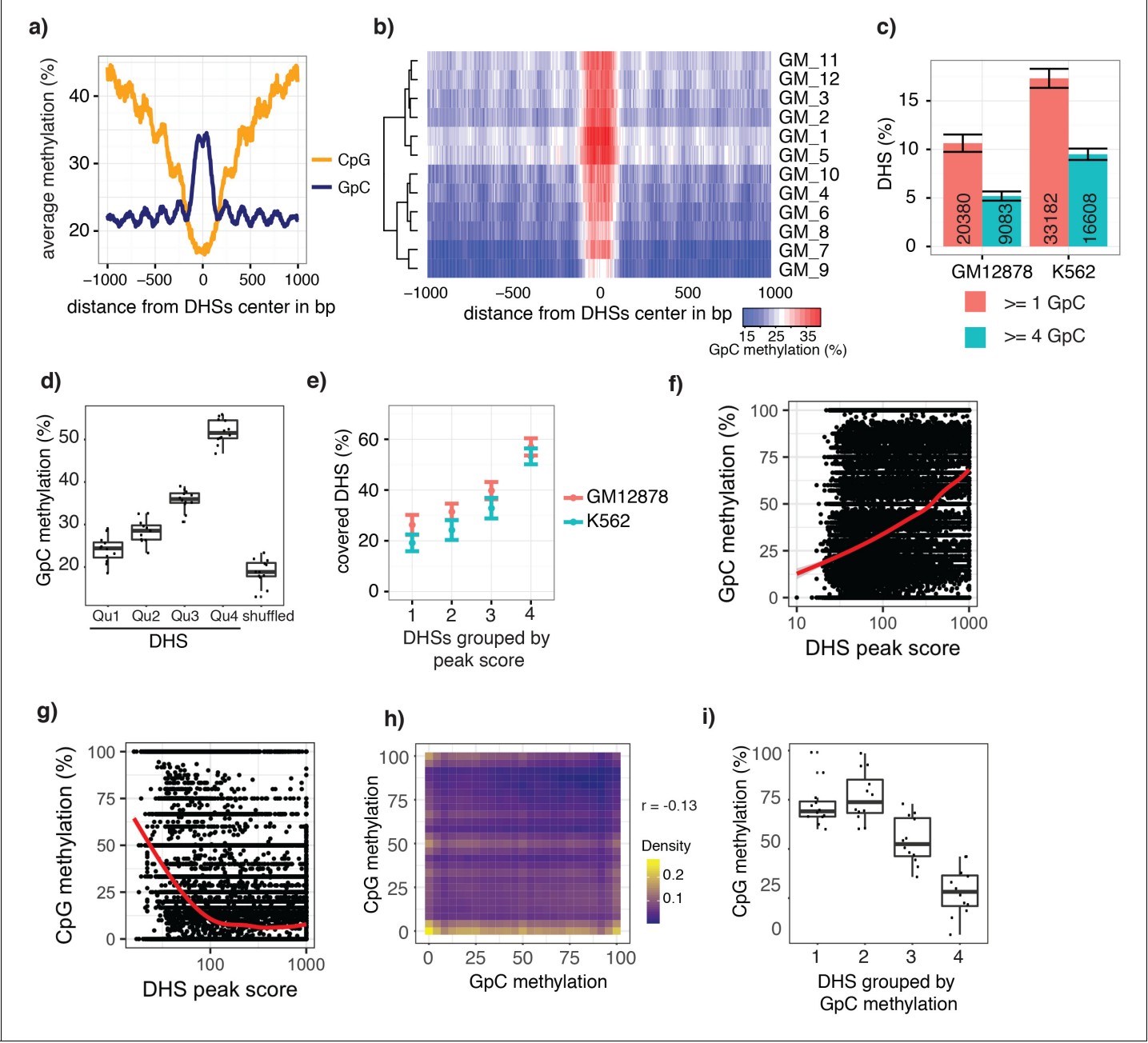

**Figure 2.** scNOMe-seq data reveal how accessibility in single cells underlies observed DNaseI hypersensitivity in a population of cells. (**a**) Average GpC methylation level (blue) and CpG methylation level (orange) at DHSs in GM12878 cells. Regions are centered on the middle of DNase-seq peak locations. Shown is the average methylation across a 2 kb window of 12 GM12878 cells. (**b**) Heatmap displaying the average GpC methylation level across the same regions as in a). Each row corresponds to an individual GM12878 cell. Cells were grouped by similarity. (**c**) Proportion of DHSs covered by scNOMe-seq sequencing reads in each cell. The proportion displayed corresponds to the fraction of DHSs covered by at least 1 or 4 GpCs in a given cell. Only DHSs with at least 1 GpC (red) or 4 GpCs (cyan) within their primary sequence were taken in consideration. Error bars represent standard deviation. (**d**) Average GpC methylation at DHSs grouped into quartiles based on associated DNase-seq peak scores from lowest to highest scores. 'Shuffled' represents methylation data in genomic regions obtained by random placements of DHS peak intervals. Data shown are from GM12878 cells. (**e**) Fraction of accessible sites in individual GM12878 cells (red) and K562 cells (cyan). Shown are the means and standard deviation based on all cells. (**f**) Scatter plot showing relationship between GpC methylation levels and DHS peaks score for each covered DHS. Plot shows data from all individual GM12878 cells. Red trend line is shown to visualize the relationship between GpC methylation and endogenous CpG methylation. (**g**) Scatter plot showing relationship between CpG methylation levels and DHS peaks score for each covered DHS. Plot shows data from all individual GM12878 cells. Red trend line is shown to visualize the relationship between CpG methylation and peak scores. (**h**) Plot illustrates the relationship between endogenous CpG methylation and GpC methylation at DHS loci. Plot shows combined data from all GM12878 cells. Correlation was

*Figure 2 continued on next page*

*Figure 2 continued*

calculated based on Pearson correlation (r = −0.13) i) Average CpG methylation at DHS loci grouped based on GpC scores within single cells. Each dot represents the average CpG methylation level for a single cell.

The following figure supplements are available for figure 2:

**Figure supplement 1.** Average CpG and GpC methylation levels in single cells.

**Figure supplement 2.** Heatmaps of average GpC and CpG methylation across DHS regions in GM12878 cells.

**Figure supplement 3.** Average GpC and CpG methylation across DHS regions in K562 cells.

**Figure supplement 4.** Heatmaps of average GpC and CpG methylation across DHS regions in K562 cells.

**Figure supplement 5.** Distribution of counts of GpCs within DHSs in GM12878 and K562 cells.

**Figure supplement 6.** Proportion of DHSs at different cutoffs for GpCs and CpGs.

**Figure supplement 7.** Relationship between DNase-seq peak score and GpC and CpG methylation in GM12878 and K562 cells.

**Figure supplement 8.** Correlation between GpC methylation and DHS peak score.

**Figure supplement 9.** Cumulative distribution of average GpC methylation in DHSs in GM12878 and K562 cells.

**Figure supplement 10.** Proportion of accessible DHSs remains stable across range of thresholds for methylation levels and covered GpCs per site.

**Figure supplement 11.** Cell-to-cell variability in DHSs accessibility reflects DNaseI hypersensitivity of the region.

**Figure supplement 12.** Comparison of correlations between single cell NOMe-seq and bulk NOMe-seq data sets.

**Figure supplement 13.** GpC methylation correlates with DHS peaks scores in individual cells.

**Figure supplement 14.** Endogenous CpG methylation is inversely correlated with DHS peak scores in individual cells.

**Figure supplement 15.** Comparison of CpG and GpC methylation status at individual DHS in single GM12878 cells.

cells, respectively (*Figure 2c*). Chromatin accessibility signal can vary along the length of a given DHSs due to binding of transcription factors (*Neph et al., 2012*) and the specific position of a GpC within a DHS will thus affect its chance of being methylated. To account for this variability and to obtain more robust estimates of GpC methylation only DHSs with at least four covered GpC were used for the subsequent analyses and referred to as 'covered DHSs'.

In single cells, the average GpC methylation at covered DHSs was strongly correlated with the observed DNaseI accessibility at these sites in bulk populations (*Figure 2d*, *Figure 2—figure supplements 7* and *8*). The opposite trend was observed for endogenous CpG methylation which was lowest for DHSs with the highest DNaseI accessibility (*Figure 2—figure supplement 7*). The correlation between GpC methylation and DNaseI accessibility was lower for scNOMe-seq data compared to bulk NOMe-seq data in the same cell line (*Figure 2—figure supplement 8*). At the level of individual sites the distribution of GpC methylation suggested that around 50% of the covered DHS showed less than 25% GpC methylation in individual cells (*Figure 2—figure supplement 9*). To estimate the proportion of covered DHSs that were concurrently accessible in a single cell I applied a fixed threshold of 40% GpC methylation above which sites were considered accessible (Materials and methods). At this GpC methylation threshold 32–44% and 26–37% of all covered DHSs were determined to be accessible in single GM12878 and K562 cells, respectively. As expected these results depended to some degree on the cutoffs used for GpC methylation and the number of required GpCs per DHS. However, even under the most lenient conditions less than 50% of DHSs

were accessible in individual cells (*Figure 2—figure supplement 10*). Grouping the DHSs based on DNaseI accessibility in bulk samples, confirmed that the degree of DNaseI accessibility related closely to the frequency of DHS accessibility in single cells (*Figure 2e*). This analysis leveraged the NOMe-seq-specific property that the DNA sequence is recovered independently of its accessibility status. It provided direct evidence for the notion that the degree of DNaseI accessibility observed in DNase-seq of bulk samples reflects the frequency with which a region is accessible in individual cells. Consequently, chromatin accessibility between cells is less variable at regions with high DNaseI accessibility in bulk samples (*Figure 2—figure supplement 11*). Correspondingly, correlation of GpC methylation between individual cells is stronger at DHS loci compared to randomized locations (*Figure 2—figure supplement 12*).

## scNOMe-seq captured characteristic chromatin organization associated with transcription

Chromatin accessibility and endogenous methylation show characteristic patterns at gene promoters and within gene bodies (*Schübeler, 2015*; *ENCODE Project Consortium, 2012*). To test whether these features can be observed in scNOMe-seq data, I first plotted the average GpC and CpG methylation around transcription start sites (TSS). The average GpC methylation showed the expected increase of chromatin accessibility directly upstream of the TSS (*Figure 3a*, *Figure 3—figure supplement 1*). In contrast, and as expected, the endogenous CpG methylation decreased towards the TSS (*Figure 3b*). To visualize the distribution of CpG methylation throughout entire gene loci, I plotted the aggregated CpG methylation across regions containing the entire gene body and 50 kb upstream and 50 kb downstream of each gene (*Figure 3c*, *Figure 3—figure supplement 1*). Endogenous methylation was specifically reduced at the narrow promoter region and gradually increased throughout the gene body. Downstream of the transcription end site (TES) the average level CpG methylation level fell back to the non-genic background level. Endogenous CpG methylation is typically increased within highly expressed genes (*Schübeler, 2015*). This trend was clearly apparent in the single cell data where gene body methylation was highest in highly expressed genes (*Figure 3d*, *Figure 3—figure supplement 1*). Correspondingly, in promoter regions (−500 bp to +150 bp) chromatin accessibility (GpC methylation) increased with the transcript level of the adjacent gene (*Figure 3e*, *Figure 3—figure supplement 2*). In contrast to chromatin accessibility, endogenous methylation was lowest in promoters of genes with high transcript levels (*Figure 3f*). These data show that scNOMe-seq recapitulated known characteristics of chromatin accessibility and endogenous methylation at gene promoters and within gene bodies.

## GpC methylation and endogenous CpG methylation data separated individual GM12878 and K562 cells

A potentially powerful application for single cell genomic approaches is the label-free classification of single cells from heterogeneous mixtures of cells solely based on the measured feature (*Cusanovich et al., 2015*; *Buenrostro et al., 2015a*; *Jaitin et al., 2014*; *Macosko et al., 2015*). Of note, using a union set of DHSs from both cell types was sufficient to classify individual GM12878 and K562 cells into their respective cell types based on GpC methylation (*Figure 4a*, *Figure 4—figure supplement 1*). While this assessment might have been influenced in part by the separate processing of the cell types, both cell types showed preferential enrichment of GpC methylation at their respective DHSs compared to DHSs identified in the other cell type (*Figure 4b*). Similar to GpC methylation, endogenous CpG methylation at multiple sets of genomic features was sufficient to separate the cells into the respective cell types (*Figure 4c*, *Figure 4—figure supplement 1*).

## Detection of footprints of CTCF binding at individual loci in single cells

To examine in detail whether scNOMe-seq captures features of chromatin accessibility that are specifically associated with transcription factor binding, I analyzed scNOMe-seq data at transcription factor binding sites (TFBS). The average GpC methylation around CTCF ChIP-seq peaks (*ENCODE Project Consortium, 2012*) in single cells recapitulated the accessibility previously observed in NOMe-seq bulk samples (*Kelly et al., 2012*): Accessibility increased strongly towards the CTCF binding sites while the location of the CTCF motif at the center of the region showed low accessibility suggesting that CTCF binding protected from GpC MTase activity and thus creating a

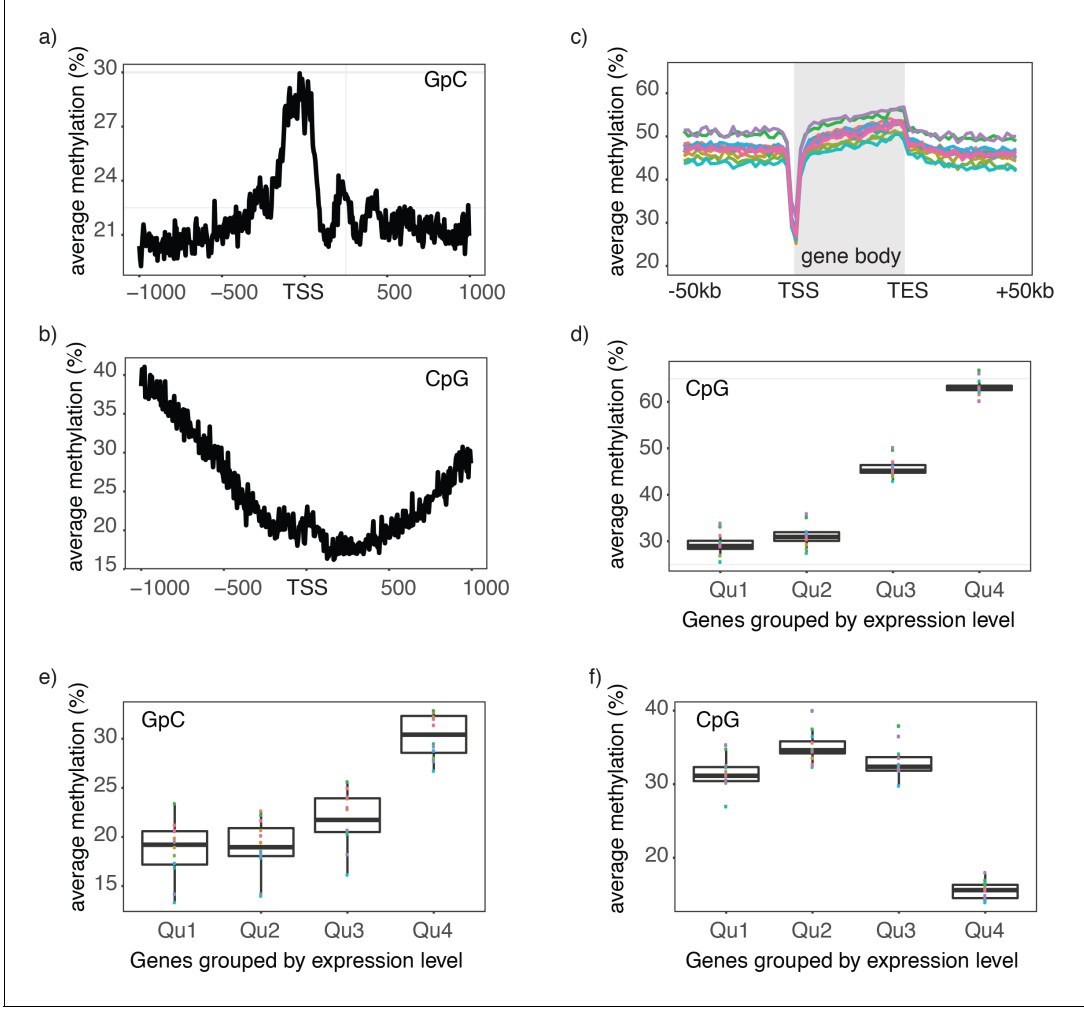

**Figure 3.** Single cell NOMe-seq reveals chromatin features closely linked to gene expression. (a) Average GpC methylation level at TSS in GM12878 cells. Regions are centered on the TSS locations. Shown is the average methylation across a 2 kb window of 12 GM12878 cells. (b) Same as in a) but displaying the endogenous CpG methylation level. (C) Average endogenous CpG methylation at gene loci in individual GM12878 cells. Shown is the average methylation across gene bodies (represented as meta genes) and 50 kb regions upstream and downstream of each gene. Each line represents the aggregated CpG methylation data for a single GM12878 cell (TES: transcription end site). (d) Boxplot displays average CpG methylation in gene bodies. Genes were grouped into quartiles based on their transcript levels in bulk. Dots represent the average CpG methylation value for individual cells. (e) Boxplot displays average GpC methylation in promoter regions (−500 bp to +150 bp). Genes were grouped into quartiles based on their transcript levels in bulk. (f) Similar to (e) but displayed are the levels of endogenous CpG methylation.

The following figure supplements are available for figure 3:

**Figure supplement 1.** Endogenous methylation in gene bodies of single K562 cells.

**Figure supplement 2.** Chromatin accessibility in promoters correlates with transcript levels of adjacent genes.

footprint of a CTCF binding event, both when averaged across data from all single cells (*Figure 5a* and *Figure 5—figure supplement 1*) and in individual cells (*Figure 5b* and *Figure 5—figure supplement 2*). In contrast, endogenous CpG methylation was generally depleted around the center of CTCF binding sites (*Figure 5a* and *Figure 5—figure supplement 1*). Similar accessibility profiles, albeit less pronounced compared to CTCF, were observed for additional transcription factors, for example EBF1 and PU.1 (*Figure 5—figure supplement 3*). These analyses provided evidence that, in aggregate, scNOMe-seq detected chromatin accessibility characteristic of CTCF binding in single cells. To test whether scNOME-seq data detected CTCF footprints at individual motifs loci, GpC

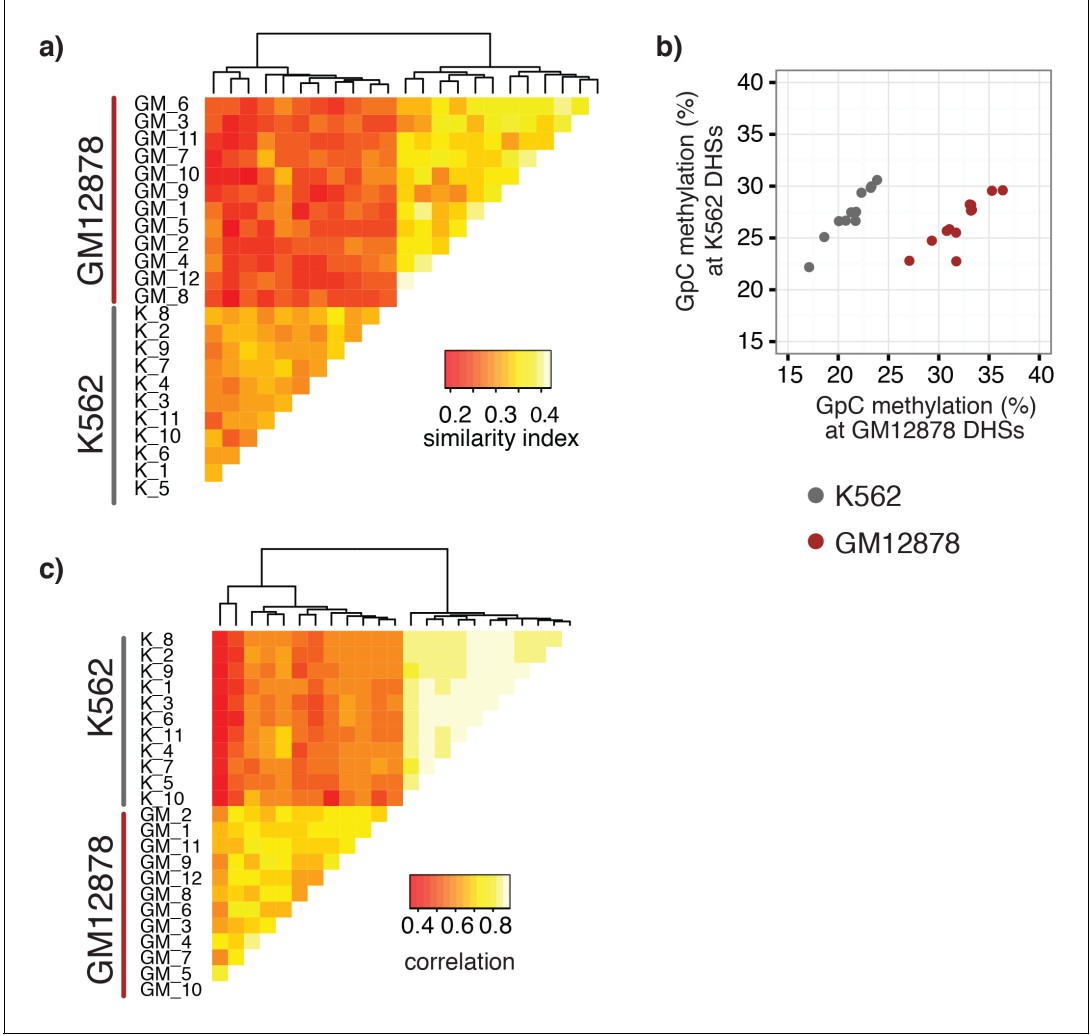

**Figure 4.** single cell GpC and CpG methylation signal is sufficient to group GM12878 and K562 cells according to their origin. (a) Heatmap shows similarity scores (pair-wise Jaccard distances) for accessibility between all GM12878 and K562 cells measured on the union set of DHSs from GM12878 and K562 cells. Cells were grouped based on unsupervised hierarchical clustering. (b) Average GpC methylation at the DHSs from GM12878 cells and K562 cells, respectively, was calculated for all individual GM12878 and K562 cells. The resulting two values for GpC methylation are displayed for each cell. While the average methylation levels at K562 DHSs for both cell types appear similar, GM12878 and K562 are separable based on these data when accounting for different levels of genome-wide GpC methylation in GM12878 and K562 cells. Importantly, for cells from either cell type the methylation levels are higher in the DHSs of the cell type of origin than in the DHSs of the other cell type. (c) Heatmap shows correlation coefficients between all GM12878 and K562 cells for pair-wise comparison of CpG methylation levels. Genome was divided into 10 kb bins and only bins with sufficient coverage in both cells were used for a given pair (> = 20 covered CpGs). Cells were grouped based on unsupervised hierarchical clustering.

The following figure supplement is available for figure 4:

**Figure supplement 1.** Single GM12878 and K562 cells can be grouped based on GpC methylation and endogenous methylation.

---

methylation at motifs within CTCF ChIP-seq peaks was compared to the GpC methylation level in the regions flanking each motif (*Figure 5c*). On average, two-thirds of CTCF motif instances within these accessible regions showed no GpC methylation, suggesting that CTCF binding prevented the GpC MTase from methylating the cytosines within the binding motif and thus creating a footprint (*Figure 5d and f*). Of note, motifs associated with a footprint had significantly higher scores than motifs without a footprint suggesting that the motif score is a strong determinant of CTCF binding within these accessible regions (p= 5.429e-12, paired t-test) (*Figure 5e,g* and *Figure 5—figure*

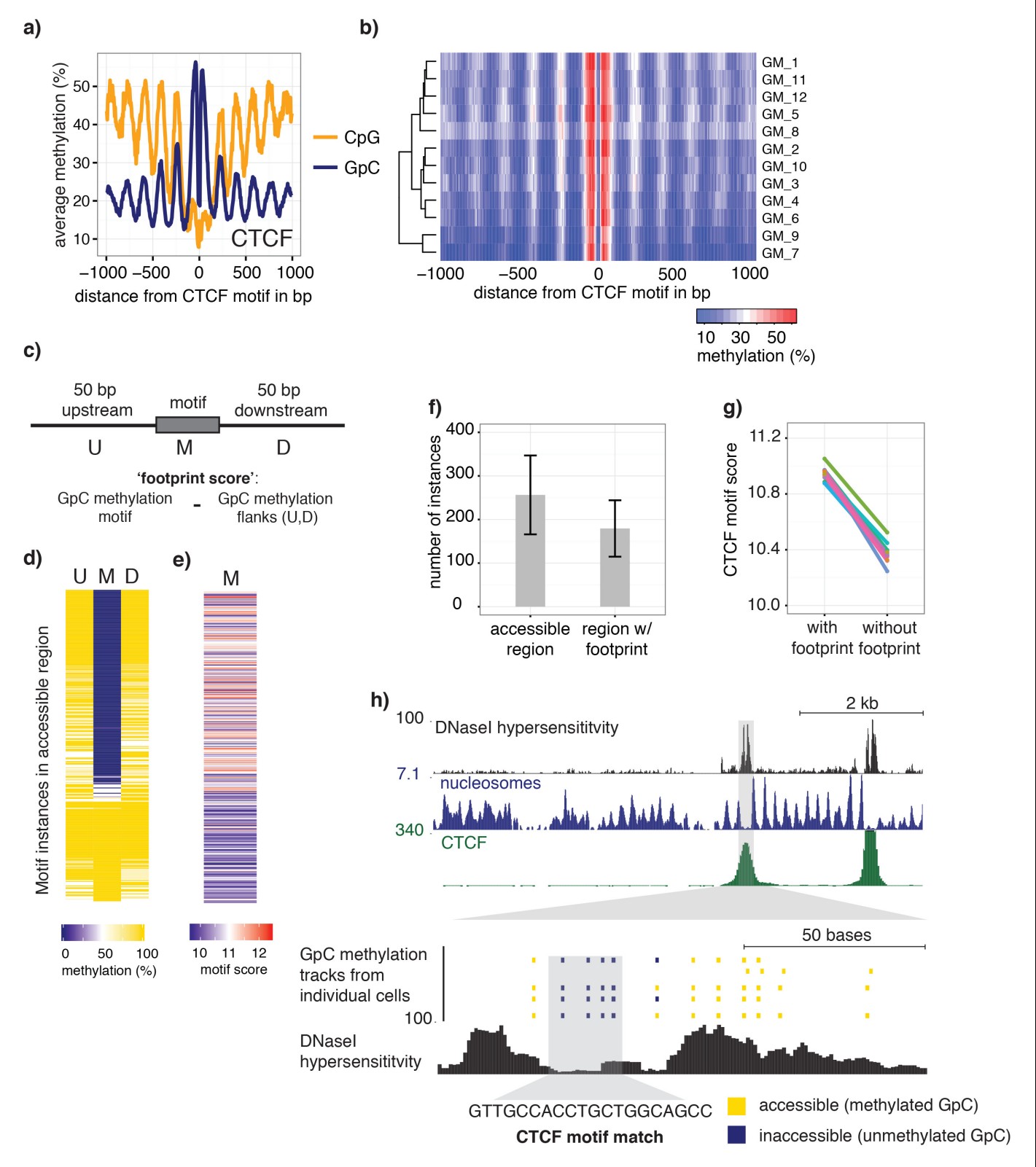

**Figure 5.** scNOMe-seq detected characteristic accessibility patterns at CTCF transcription factor binding sites and measured CTCF footprints at individual loci. (a) Average GpC methylation level (blue) and CpG methylation level (orange) at CTCF binding sites in GM12878 cells. Regions are centered on motif locations. Shown is the average methylation across a 2 kb window of the pool of 12 GM12878 cells. (b) Heatmap displaying the average GpC methylation across CTCF binding sites. Each row corresponds to an individual GM12878 cell and rows are grouped by similarity. (c)
*Figure 5 continued on next page*

*Figure 5 continued*

Schematic outline the measurement of CTCF footprints in accessible regions. M denotes CTCF binding motifs within CTCF ChIP-seq regions and U and D indicate 50 bp upstream and downstream flanking regions. footprint score was determined by subtracting the average GpC methylation in the flanking regions from the GpC methylation at the motif. (d) Heatmap displays GpC methylation in accessible regions found in a representative GM12878 cell (GM_1). Each row represents a single CTCF motif instance within a CTCF ChIP-seq region. Average methylation values for the motif and the 50 bp upstream and downstream regions are shown separately. Regions are sorted based on the footprint score. Displayed are only regions that had sufficient GpC coverage and that were considered accessible based on the methylation status of the flanking regions. (e) Heatmap reporting the CTCF motif scores for the motif regions in (d). Regions are sorted in the same order as in (d). (f) Average number of accessible regions at CTCF motifs and the average number of those with a detectable footprint per individual GM12878 cell. Error bars reflect standard deviation. (g) Average CTCF motif scores in regions with and without CTCF footprint for all 12 GM12878 cells. Each line connects the two data points from an individual cell. Regions with footprint are associated with higher motif scores (p= 5.429e-12, paired t-test). (h) Combined display of scNOMe-seq data from this study and DNase hypersensitivity data, nucleosome occupancy, and CTCF ChIP-seq data from ENCODE. Upper panel shows a ~10 kb region containing a CTCF binding site. DNaseI hypersensitivity data and nucleosome density show characteristic distribution around CTCF binding sites in GM12878 cells. Lower panel shows the GpC methylation data of 5 individual cells that had sequencing coverage in this region, 4 of the cells provide GpC data covering the CTCF motif located in the region. scNOMe-seq data tracks show methylation status of individual GpCs. Each row corresponds to data from a single cell. These data indicate that binding of CTCF is detected in all 4 cells. Data are displayed as tracks in the UCSC genome browser (http://genome.ucsc.edu).

The following figure supplements are available for figure 5:

**Figure supplement 1.** Average GpC methylation and endogenous CpG methylation at CTCF sites in pooled K562 cells.

**Figure supplement 2.** Average GpC methylation level at CTCF binding sites in individual K562 cells.

**Figure supplement 3.** Average GpC methylation and endogenous CpG methylation at additional transcription factor binding sites in pools of GM12878 and K562 cells.

**Figure supplement 4.** Scores at CTCF motifs with footprints are significantly higher than those without.

**Figure supplement 5.** Loci with CTCF footprint in single cells.

*supplement 4*). Of note, the CTCF footprints could be observed at individual loci within individual cells and were shared across cells (*Figure 5h* and *Figure 5—figure supplement 5*).

## Estimating nucleosome phasing in single cells

The pattern of GpC methylation adjacent to CTCF sites suggested that scNOMe-seq also detected the well-positioned nucleosomes flanking these regions (*Figure 5a*) (*Kelly et al., 2012*). This observation was confirmed by the oscillatory distribution of the average GpC and CpG methylation around locations of well-positioned nucleosomes identified from MNase-seq data (*ENCODE Project Consortium, 2012*) (*Figure 6a*). While nucleosome core particles are invariably associated with DNA fragments of 147 bp, nucleosomes are separated by linker DNA of varying lengths, resulting in different packaging densities between cell types and between genomic regions within a cell (*Valouev et al., 2011*; *Schones et al., 2008*). To determine whether scNOMe-seq data can be used to measure the average linker length, average distances between nucleosome midpoints in single cells (phasing distances) were estimated by correlating the methylation status between pairs of cytosines in GpC di-nucleotides at offset distances from 3 bp to 400 bp (*Figure 6c,d* and *Figure 6—figure supplements 1* and *2*). The estimated phases fell between 187 bp and 196 bp (mean = 196.7 bp) in GM12878 cells, and between 188 bp and 200 bp (mean = 194.2 bp) in K562 cells (*Figure 6e*). These estimates are in general agreement with phase estimates derived from MNase-seq data in human cells (*Valouev et al., 2011*). In addition, estimated phasing distances varied within individual cells depending on the chromatin context, similar to observation from bulk MNase-seq data (*Valouev et al., 2011*) (*Figure 6f*).

## Discussion

In this study, I demonstrated that scNOMe-seq simultaneously measures chromatin accessibility by GpC methylation as well as endogenous CpG and DNA methylation in single cells. scNOMe-seq

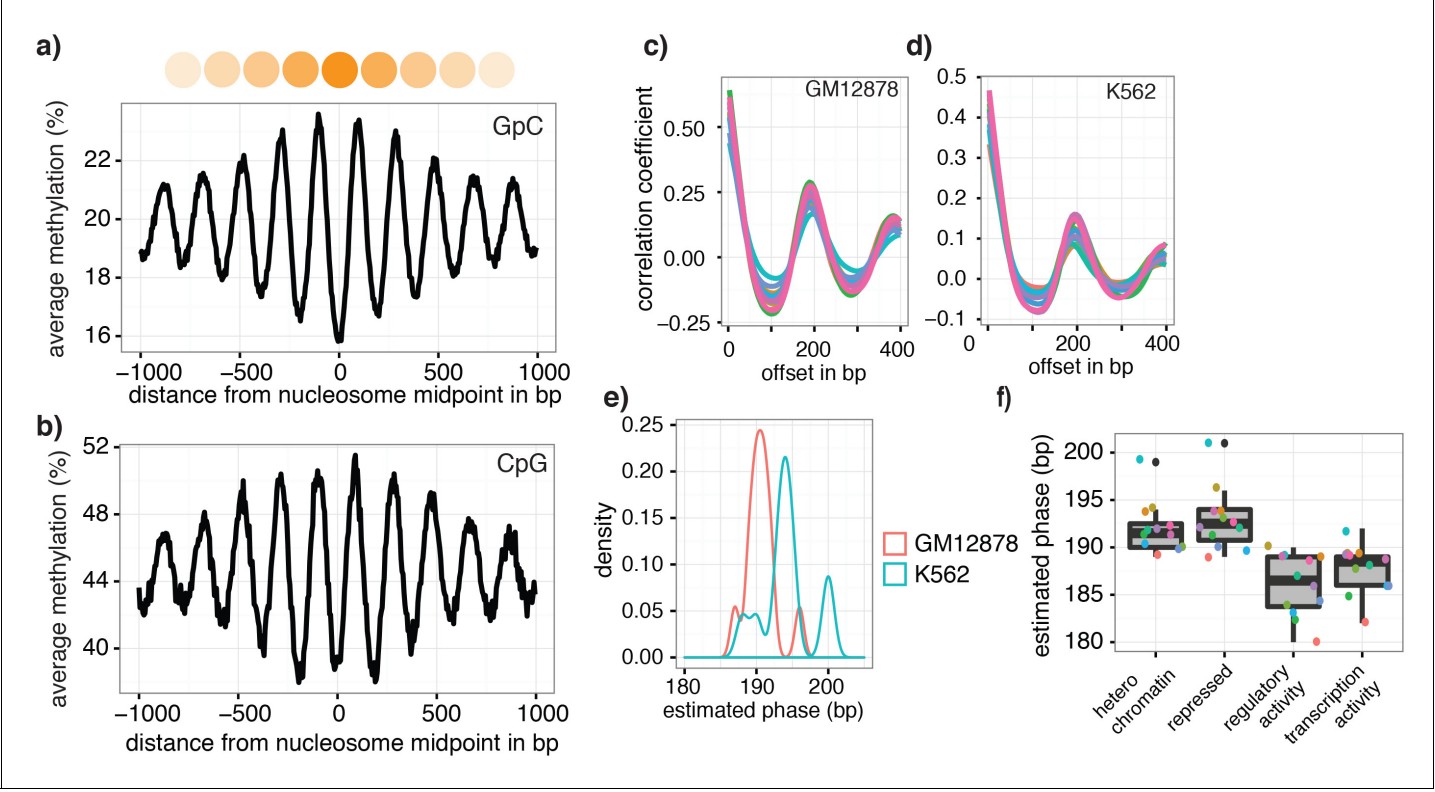

**Figure 6.** Nucleosome phasing in single cells. (a) Average GpC methylation level and (b) CpG methylation level at well-positioned nucleosomes in GM12878 cells. Regions are centered on midpoints of top 5% of positioned nucleosomes. Shown is the average methylation across a 2 kb window of the pool of 12 GM12878 cells. (c), (d) Correlation coefficients for the comparison in methylation status between GpCs separated by different offset distances for GM12878 (c) and K562 (d) cells. Each line represents a single cell. Data are smoothened for better visualization. (e) Distribution of estimated phase lengths for GM12878 and K562 cells. (f) Nucleosome phasing in GM12878 in genomic regions associated with different chromatin states defined by chromHMM (ENCODE). Boxplot represents the distribution of estimated phase lengths from all 12 GM12878 cells and overlaid points indicate values of each individual cells.

The following figure supplements are available for figure 6:

**Figure supplement 1.** Number of nucleotide pairs used for correlation at each offset distance.

**Figure supplement 2.** Offset distances with a high proportion of GpC pairs that share methylation status indicate average phasing distance.

detected chromatin accessibility at DHSs and TFBS and, in aggregate, these data recapitulated NOMe-seq data obtained from bulk cells (*Kelly et al., 2012*). scNOMe-seq data also detected footprints of CTCF binding, and was used to estimate nucleosome phasing distances.

Similar to other single cell genomic methods, scNOMe-seq relies on annotations obtained from bulk measurements (*Cusanovich et al., 2015*; *Buenrostro et al., 2015b*; *Smallwood et al., 2014*; *Farlik et al., 2015*). A limitation of single cell genomic methods is their sparse coverage which leads to high allelic drop-out. For methods in which the signal is based on counting the sequenced fragments, such as ATAC-seq and DNase-seq, this poses a challenge since true negatives at a specific location cannot be distinguished from false negatives that are a consequence of read loss. Compared to these methods, scNOMe-seq has the unique advantage, that reads are recovered independently of the signal and allelic drop-out events therefore can be distinguished from closed or inaccessible chromatin configurations. The frequency of accessible sites in the population of DHSs can be estimated. Using this approach only about 30–50% of DHSs detected in the population were found accessible in a single cell, depending on the thresholds chosen to call a site accessible. While this assessment would have been possible using bulk NOMe-seq data, scNOMe-seq offers important

possibilities for future applications. For example, to compare accessibility across multiple loci within a single cell and the use of heterogeneous cellular mixtures as input material.

As expected, the chance of a covered DHS to being open or closed is not equally distributed across all DHSs from the population. Instead, DHSs with strong DNaseI accessibility showed a higher frequency of accessibility in single cells compared to those sites with low DNaseI accessibility in the population (*Figure 2e*) suggesting that the peak height is indeed directly related to the frequency with which a site is accessible in individual cells. In agreement with this observation a large proportion of variability observed between cells was attributable to DHSs with low DNaseI accessibility in bulk samples (*Figure 2—figure supplement 11*). In principle, variation between cells could be due to differential GpC MTase enzyme activity. However, the genome-wide levels of GpC methylation reached comparable levels in all cells and the variability between cells was not equally distributed across all DHS (*Figure 2d*, *Figure 2—figure supplement 1*)

Measuring similarity of chromatin accessibility between cells was sufficient to group GM12878 and K562 cells based on their cell type of origin (*Figure 3a*). In this particular case, the separation is confounded with experimental batches. However, higher average GpC methylation in DHSs for the respective cell type compared to the DHSs of the other cell type indicated that scNOMe-seq can differentiate the two cell types (*Figure 2—figure supplement 14*). Similarly, endogenous CpG methylation at different genomic features (DHS, 10 kb windows, gene bodies) was sufficient to distinguish between the two cell types. This approach should be extendable to scNOMe-seq data from samples containing mixtures of cell types.

scNOMe-seq measures chromatin accessibility at GpC di-nucleotides along the entire length of a sequencing read. Since most features that bind DNA are smaller than the length of 100 bp (200 bp within 200–500 bp regions in the case of paired end reads), the regions covered by sequence-specific transcription factors and nucleosomes can be captured within a single fragment. This allows one to directly detect binding of TFs provided that their sequencing motif contains at least one GpC dinucleotide. I demonstrated the feasibility of this approach using CTCF binding sites. Of note, most motifs within regions of CTCF ChIP-seq peaks were protected from GpC methylation ('footprint') (*Figure 5*). In agreement with an inferred binding event as the cause for this protection, scores for CTCF motifs that were associated with a footprint were significantly higher than for motifs without a footprint. Depending on the motif specificity of a given TF and provided that their motifs contain a GpC dinucleotide, similar measurements should be feasible for many TFs and could be used to infer the activity of a range of transcription factors in single cells or to measure combinatorial binding of two or more TFs.

Estimation of the average nucleosome phasing distances allows one to study chromatin compaction and complements the measurements of chromatin accessibility at regulatory regions and DNA methylation. The estimates from individual cells fit very well with measurements made from MNase-seq data in bulk samples (*Valouev et al., 2011*). It remains to be established whether the variation in phasing distances between individual cells is of biological or technical nature (*Figure 6e*).

These proof-of-principle experiments have been performed using commercial kits for bisulfite conversion and library amplification, additional optimization or alternative amplification approaches (*Smallwood et al., 2014*) are likely to increase the yield substantially. Compared to other single cell methods, for example ATAC-seq, scNOMe-seq does not enrich for accessible chromatin regions and thus requires significantly more sequencing coverage. Ultimately, it should be possible to integrate the GpC MTase treatment into microfluidic workflows and combine this method with scRNA-seq, similar to recently published methods that combine scRNA-seq and methylome- sequencing (*Angermueller et al., 2016*). This study was primarily designed to test the feasibility of NOMe-seq in single cells and only a small number of nuclei where sequenced for each cell line. As a consequence, this set up could not be used to study cell-to-cell variation in detail. scNOMe-seq will be particularly useful for studies that aim to simultaneously measure chromatin accessibility and DNA methylation. This approach will be especially powerful for the characterization of chromatin organization in single cells from heterogeneous mixtures or complex tissues, for example to samples of brain tissues or primary cancer cells.

## Materials and methods

### Cell culture, nuclei isolation, and GpC methylase treatment

GM12878 (RRID:CVCL_7526) and K562 (RRID:CVCL_0004) cells were obtained directly from Coriell and ATCC, respectively. No further confirmation of the authenticity of these cell lines or mycoplasma testing has been performed. GM12878 were grown in RPMI medium 1640 (Gibco), supplemented with 2 mM L-Glutamine (Gibco), and Penicilin and Streptavidin (Pen Strep, Gibco), and 15% fetal bovine serum (FBS, Gibco). K562 were grown in RPMI medium 1640 of the same composition but with 10% FBS. Cells were grown at 37 C and in 5% CO2. NOMe-Seq procedure was performed based on protocols for CpG methyltransferase M.SSsI described in *Miranda et al. (2010)* and the GpC methyltransferase from *M.CviPI* (*Kelly et al., 2012*), with some modification. Between 2 × 106 and 5 × 10^6 cells were harvested by centrifuging the cell suspension for 5 min at 500x g. Cells were washed once with 1x PBS, re-suspended in 1 ml lysis buffer (10 mM Tris-HCl pH 7.4, 10 mM NaCl, 3 mM MgCl2) and incubated for 10 min on ice. IGEPAL CA-630 (Sigma) was added to a final concentration of 0.025% and the cell suspension was transferred to a 2 ml Dounce homogenizer. Nuclei were released by 15 strokes with the pestle. Success of lysis was confirmed by inspection under a light microscope. Nuclei were collected by centrifuging the cell suspension for 5 min at 800x g at 4 C and washed twice with cold lysis buffer without detergent. One million nuclei were resupended in reaction buffer to yield a suspension with a final concentration of 1x GpC MTase buffer (NEB), 0.32 mM S-Adenosylmethionine (SAM) (NEB), and 50 ul of GpC methyltransferase (4 U/ul)) from *M.CviPI* (NEB). The final reaction volume was 150 ul. The suspension was carefully mixed before incubating for 8 min at 37 C after which another 25 ul of enzyme and 0.7 ul of 32 mM SAM were added for an additional 8 min incubation at 37C. To avoid disruption of nuclei incubation was stopped by adding 750 ul of 1x PBS and collecting the nuclei at 800 xg. Supernatant was removed and nuclei were re-suspended in 500 ul 1x PBS containing Hoechst 33342 DNA dye (NucBlue Live reagent, Hoechst). Nuclei were kept on ice until sorting. For preparation of bulk libraries in GM21878 cell, nuclei preparation and GpC MTase treatment was performed as described above. Nuclei were lysed immediately after incubation and DNA was isolated using Phenol/Chloroform purification.

### Nuclei isolation using fluorescence activated cell sorting (FACS), lysis, and DNA bisulfite conversion

Nuclei were sorted at the Flow Cytometry core at the University of Chicago on a BD FACSAria or BD FACSAria Fusio equipped with a 96-well-plate holder. To obtain individual and intact nuclei gates were set on forward and side scatter to exclude aggregates and debris. DAPI/PacBlue channel or Violet 450/500 channel were usedto excited the Hoechst 33342 DNA dye and to gate on cells with DNA content corresponding to cells in G1 phase of the cell cycle in order to maintain similar DNA content per cell and to remove potential heterogeneity attributable to cell cycle. Cells were sorted into individual wells pre-filled with 19 ul of 1x M-Digestion buffer (EZ DNA Methylation Direct Kit, Zymo Research) containing 1 mg/ml Proteinase K. Following collection, the plates were briefly spun to collect droplets that might formed during handling. Nuclei were lysed by incubating the samples at 50 C for 20 min in a PCR cycler. DNA was subjected to bisulfite conversion by adding 130 ul of freshly prepared CT Conversion reagent (EZ DNA Methylation Direct Kit, Zymo) to the lysed nuclei. Conversion was performed by denaturing the DNA at 98 C for 8 min followed by 3.5 hr incubation at 65 C. DNA isolation was performed using the EZ DNA Methylation Direct Kit (Zymo Research) following the manufacturer's instruction with the modification that the DNA was eluted in only 8 ul of elution buffer.

### Library preparation and sequencing

Libraries were prepared using the Pico Methyl-seq Library prep Kit (Zymo Research) following the manufacturer's instruction for low input samples. Specifically, the random primers were diluted 1:2 before the initial pre-amplification step and the first amplification was extended to a total of 10 amplification cycles. Libraries were amplified with barcoded primers allowing for multiplexing. The sequences can be found in *Supplementary file 1*, primers were ordered from IDT. The purification of amplified libraries was performed using Agencourt AMPureXP beads (BeckmannCoulter), using a

1:1 ratio of beads and libraries. Concentration and size distribution of the final libraries was assessed on an Bioanalyzer (Agilent). Libraries with average fragment size above 150 bp were pooled and sequenced. Libraries were sequenced on Illumina HiSeq 2500 in rapid mode (K562 cells) and HiSeq4000 (GM12878 cells).

## Read processing and alignment

Sequences were obtained using 100 bp paired-end mode. For processing and alignment each read from a read pair was treated independently as this slightly improved the mapping efficiency. Before alignment, read sequences in fastq format were assessed for quality using fastqc (http://www.bioin-formatics.babraham.ac.uk/projects/fastqc/). Reads were trimmed to remove low quality bases and 6 bp were clipped from the five prime end of each read to avoid mismatches introduced by amplification. In the case of GM12878 cells 6 bp were clipped from either end of the read. Only reads that remained longer than 20 bp were kept for further analyses. These processing steps were performed using trim_galore version 0.4.0 (http://www.bioinformatics.babraham.ac.uk/projects/trim_galore/) with the following settings: *trim_galore –quality 30 –phred33 –illumina –stringency 1 -e 0.1 –clip_R1 6 –gzip –length 20 – output_dir outdir Sample.fastq.gz*. The trimmed fastq files were aligned using the bisulfite aligner bismarck version 0.15.0 (*Krueger et al., 2012*) which calls bowtie2 (*Langmead and Salzberg, 2012*) internally. Reads were aligned to the human genome (genome assembly hg38). Reads were aligned in single read mode using default settings. The amplification protocol used to generate the scNOMe-seq libraries yielded non-directional libraries and alignment was performed with the option —non_directional (*bismark –fastq –prefix SamplePrefix –output_dir output_dir – non_directional –phred33-quals –score_min L,0,–0.2 –bowtie2 genome_file trimmed. fastq.gz*). Some libraries contained small amounts of DNA from *C. elegans* as spike-ins, however these were not used during the analysis. Duplicates were removed using samtools version 0.1.19 (*Li et al., 2009*) on sorted output files from bismark (*samtools rmdup Sample Prefix.sorted.bam Sample Aligned_rmdup.bam*).

## Extraction of GpC and CpG methylation status

Coverage and methylation status of all cytosines was extracted using bismark_methylation_extractor (*Krueger et al., 2012*) (*bismark_methylation_extractor -s –ignore 6 –output outdir –cytosine_report –CX –genome_folder path_to_genome_data SampleAligned_rmdup.bam*). The resulting coverage files were used to extract the methylation status of cytosines specifically in GpC and CpG di-nucleo-tides using the coverage2cytosine script which is part of Bismark (*Krueger et al., 2012*). The result-ing coverage files contained cytosines in GCG context which are ambiguous given that they represent a cytosine both in GpC and CpG di- nucleotides. Coordinates of these ambiguous posi-tions were identified using oligoMatch (*Kent et al., 2002*) and these positions were removed from the coverage files. The number of unconverted cytosines (estimated based on apparent methylation rates in non-GpC and non-CpG context) was low in all libraries (<1%). However, it was noted that unconverted cytosines were not randomly distributed but associated with entirely unconverted reads. Regions covered by a read with more than three unconverted cytosines in non-CpG and non-GpC context were removed from further analysis as well. The genotype was not taken into account as its effect on calling the methylation status incorrectly was deemed negligible for the analyses per-formed here.

## Analysis of GpC and CpG methylation at genomic features in single cells

ScNOMe-seq data were compared to a number of genomic features in GM12878 and K562 cells col-lected by Encode (*ENCODE Project Consortium, 2012*) which were downloaded through the UCSC data repository (*Karolchik et al., 2014*). These datasets are listed in *Supplementary file 1*. While the scNOMe-seq data were aligned against human genome assembly hg38, some of the datasets were only available on genome assembly hg19 and the coordinates of these datasets were lifted from hg19 to hg38 using liftOver (*Kent et al., 2002*) (default re-mapping ratio 0.95). Nucleosome positions based on MNase-seq data in GM12878 were determined with DANPOS version 2.2.2 (*Chen et al., 2013*) using default settings. Resulting intervals were lifted to hg38. After removing

summit locations with occupancy values above 300, the top 5% (713361) of nucleosome positions based on their summit occupancy value were used.

GpC and CpG methylation density across intervals encompassing DNase hypersensitivity sites (DHSs), transcription factor binding sites (TFBS), and well positioned nucleosomes was calculated across the 2 kb regions centered on the middle of these regions using the scoreMatrixBin function in the genomation package (*Akalin et al., 2015*) in R (*R Core Team, 2015*). Data were aggregated in 5 bp bins for each region and across all regions covered in a single cell. The average methylation level in pre-defined intervals (DHSs, TFBS) was determined by computing the average GpC or CpG methylation for each interval together with the number of GpC/CpGs covered in this interval using the map function in bedtools (*Quinlan and Hall, 2010*). If no other cut-offs were given, DHSs were considered 'covered' and used in analyses when at least 4 GpCs occurring within the predefined interval were covered by sequencing data in an individual cell. Because the frequency of CpG di-nucleotides is significantly lower, only 2 CpGs were required in order for a DHSs to be considered covered for analyses that focused on endogenous DNA methylation. To count the number of cytosines within the primary sequence of a given DHSs only cytosines on the forward strand were counted. While each GpC dinucleotide can be measured on both strands and would therefore yield a count of two cytosines the data are sparse and each location will get at most a single read. This approach should therefore give a more conservative estimate of the possible GpC coverage. For analyses that used the scores of the peak regions, the peak scores reported the datasets from bulk samples were used (*ENCODE Project Consortium, 2012*).

For analyses that were centered on transcription factor binding motifs the PWMs were obtained from the JASPAR database (*Tan, 2014*) (Tan) for the TFs CTCF (MA0139), EBF1 (MA0154), and PU.1 (MA0080). Genome-wide scanning for locations of sequence matches to the PWMs was performed using matchPWM in the Biotstring package (*Pages et al., 2016*) in R with a threshold of 75% based on the human genome assembly hg38.

All plots were prepared using ggplot2 (*Wickham, 2009*), with the exception of heatmaps displaying the average methylation density around genomic features in individual cells which were prepared using heatmap.2 in gplots (*Warnes et al., 2016*).

## Comparison of chromatin accessibility between cells

Similarity in accessible chromatin between cells was calculated based on Jaccard similarity. Jaccard similarity index (*Equation 1*) was calculated between pairs of samples by first obtaining the intersection of DHSs covered in both samples of a pair with more than 4 GpCs. Each feature was annotated as open or closed, depending on the methylation status (> = 40% methylation) and only pairs in which at least one of the members was open were considered for this comparison.

$$jac(A,B) = \frac{(A \cap B)}{(A \cup B)} \tag{1}$$

The similarity between samples from GM12878 and K562 cells was calculated based on the union set of DHSs from both cell lines. The similarity indexes of all pairwise comparisons were used to compute the distances between each cell. The resulting clustered data were displayed as a heat map.

## CTCF footprints in single cells

CTCF footprints were measured by comparing the GpC methylation level in each motif to the methylation level in the 50 bp flanking regions immediately upstream and downstream of the motif. Overlapping motifs were merged into a single interval before determining the coordinates for flanking regions. To ensure sufficient GpC coverage for each interval the resulting three adjacent intervals for each locus were required to contain at least one covered GpC each, and four covered GpCs in total. This analysis only included regions that were accessible based on the methylation status of the flanking regions (at least 50%). A CTCF 'footprint score' was determined by simply subtracting the average GpC methylation of the flanking regions from the GpC methylation of the motif.

scNOMe-seq data were displayed in the UCSC genome browser (*Kent et al., 2002*) by converting the GpC methylation coverage file into a bed file and using the methylation value as score. To facilitated the visualization of the data in the context of previous Encode data the methylation files were lifted to hg19. The tracks shown together with scNOMe-seq data are Open Chromatin by

DNaseI HS from ENCODE/OpenChrom (Duke University) for DNaseI hypersensitivity, Nucleosome Signal from ENCODE/Stanford/BYU, and CTCF ChIP-seq signal from Broad Histone Modification by ChIP-seq from ENCODE/Broad Institute. All data are from GM12878 cells.

### Estimation of nucleosome phasing

Nucleosome phasing estimates were obtained by first calculating the correlation coefficients for the methylation status of pairs of GpCs ad different offset distances. These values were computed using a custom python script (*Source code 1* – NucPhasing.py). Essentially, pairs of sequenced cytosines in GpC di-nucleotides were collected for each offset distance from 3 bp to 400 bp cytosine. At each offset distance the correlation of the methylation status was calculated across all pairs. Correlation coefficients were plotted against the offset distances revealing periodic changes in the correlation coefficient. The smoothened data were used to estimate the phasing distances by obtaining the off-set distance corresponding to the local maximum found between 100 bp and 300 bp. To determine phase lengths of nucleosomes in different chromatin contexts the GpC coverage files were filtered for positions falling into categories defined by chromHMM (*ENCODE Project Consortium, 2012*; *Ernst et al., 2011*) before obtaining the correlation coefficients.

### Data access

Raw data and methylation coverage files are available at GEO (https://www.ncbi.nlm.nih.gov/geo/) under the accession number GSE83882.

## Acknowledgements

I would like to thank Yoav Gilad for support, and Greg Crawford and colleagues in the Department of Human Genetics for helpful suggestions and comments on the manuscript. Cell sorting was performed by M. Olson and D. Leclerc at the Flow Cytometry core of the University of Chicago. I am grateful to Jason Lieb for his input and support at the beginning of this project.

## Additional information

### Funding
The author declares that there was no funding for this work.

### Author contributions
SP, Conceptualization, Investigation, Visualization, Writing—original draft, Writing—review and editing

### Author ORCIDs
Sebastian Pott, http://orcid.org/0000-0002-4118-6150

## Additional files

### Supplementary files
• Supplementary file 1. Table 1: scNOMe-seq libraries used in this paper and their technical details and alignment summary statistics. Table 2: Primer sequences of primers used for amplification and barcoding of sequencing library. Table 3: Additional datasets used in this study and their sources.

• Source code 1. Script to calculate autocorrelation between GpCs at offset distances from 3–400bp.

### Major datasets
The following datasets were generated:

| Author(s) | Year | Dataset title | Dataset URL | Database, license, and accessibility information |
|---|---|---|---|---|
| Pott S | 2016 | Simultaneous measurement of chromatin accessibility, DNA methylation, and nucleosome phasing in single cells | https://www.ncbi.nlm.nih.gov/geo/query/acc.cgi?acc=GSE83882 | Publicly available at the NCBIGene Expression Omnibus (accession no: GSE83882) |

The following previously published datasets were used:

| Author(s) | Year | Dataset title | Dataset URL | Database, license, and accessibility information |
|---|---|---|---|---|
| The ENCODE Project Consortium | 2012 | DNaseI HS peaks; GM12878 cells | http://hgdownload.cse.ucsc.edu/goldenPath/hg38/database/ | Publicly available under link provided (file name: wgEncodeRegDnaseUwGm12878Peak) |
| The ENCODE Project Consortium | 2012 | DNaseI HS peaks; K562 cells | http://hgdownload.cse.ucsc.edu/goldenPath/hg38/database/ | Publicly available under link provided (file name: wgEncodeRegDnaseUwK562Peak) |
| The ENCODE Project Consortium | 2012 | CTCF ChIP-seq; GM12878 cells | http://hgdownload.cse.ucsc.edu/goldenPath/hg19/encodeDCC/wgEncodeAwgTfbsUniform/ | Publicly available under link provided (file name: wgEncodeAwgTfbsBroadGm12878CtcfUniPk.narrowPeak.gz) |
| The ENCODE Project Consortium | 2012 | CTCF ChIP-seq; K562 cells | http://hgdownload.cse.ucsc.edu/goldenPath/hg19/encodeDCC/wgEncodeAwgTfbsUniform/ | Publicly available under link provided (file name: wgEncodeAwgTfbsBroadK562CtcfUniPk.narrowPeak.gz) |
| The ENCODE Project Consortium | 2012 | Pu.1 ChIP-seq; GM12878 cells | http://hgdownload.cse.ucsc.edu/goldenPath/hg19/encodeDCC/wgEncodeAwgTfbsUniform/ | Publicly available under link provided (file name: wgEncodeAwgTfbsHaibGm12878Pu1Pcr1xUniPk.narrowPeak.gz) |
| The ENCODE Project Consortium | 2012 | Pu.1 ChIP-seq; K562 cells | http://hgdownload.cse.ucsc.edu/goldenPath/hg19/encodeDCC/wgEncodeAwgTfbsUniform/ | Publicly available under link provided (file name: wgEncodeAwgTfbsHaibK562Pu1Pcr1xUniPk.narrowPeak.gz) |
| The ENCODE Project Consortium | 2012 | EBF1 ChIP-seq; GM12878 cells | http://hgdownload.cse.ucsc.edu/goldenPath/hg19/encodeDCC/wgEncodeAwgTfbsUniform/ | Publicly available under link provided (file name: wgEncodeAwgTfbsHaibGm12878Ebf1sc137065Pcr1xUniPk.narrowPeak.gz) |
| The ENCODE Project Consortium | 2012 | MNase-seq; GM12878 | http://hgdownload.cse.ucsc.edu/goldenPath/hg19/encodeDCC/wgEncodeSydhNsome/ | aligned Bam files publicly available under link provided |

| Lay FD | 2015 | NOMe-seq; K562 | https://www.ncbi.nlm.nih.gov/geo/query/acc.cgi?acc=GSM1583563 | Publicly available at the NCBIGene Expression Omnibus (accession no: GSM1583563) |
| Lay FD | 2015 | NOMe-seq; K562 | https://www.ncbi.nlm.nih.gov/geo/query/acc.cgi?acc=GSM1583567 | Publicly available at the NCBIGene Expression Omnibus (accession no: GSM1583567) |

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
