## [Decision Letter]

Thank you for submitting your article "Simultaneous measurement of chromatin accessibility, DNA methylation, and nucleosome phasing in single cells" for consideration by *eLife*. Your article has been favorably evaluated by Aviv Regev (Senior Editor) and three reviewers, one of whom, Bing Ren (Reviewer #1), is a member of our Board of Reviewing Editors.

The reviewers have discussed the reviews with one another and the Reviewing Editor has drafted this decision to help you prepare a revised submission.

Summary:

This manuscript presents a novel genomic technique single-cell NOMe-seq that allows the simultaneous measurement of DNA methylation and chromatin accessibility. Applying scNOMe-seq to GM12878 and K562 cell lines, the author provides evidence that scNOMe-seq can potentially distinguish between cell types, reveal chromatin accessibility and TF binding events, and inform about nucleosome phasing in single cells.

While all three reviewers recognize the novelty of the method, they all raised concern regarding the utilities of scNOMe-seq, and lack of novel biological insights. To strengthen the work, the author would need to carry out in depth analysis of the datasets, and expand the analyses to additional cells. The essential revisions are comprehensive and may take some time to complete. Please let us know how you plan to respond to the concerns below and provide an estimate of the time it will take to do so.

Essential revisions:

1) Much of the analyses presented in this manuscript are meta-analyses, where either multiple cells are grouped together or similar genomic features are grouped together. While such analysis is important to show that the data are consistent with bulk NOMe-seq, it does not convey the true utility of single-cell NOMe-seq. To demonstrate the utility of this method over state of art single cell technique, the author needs to carry out additional analysis, such as identifying accessible chromatin regions from scNOMe-seq data, and carrying out systematic comparison to existing methods.

2) Better characterize the cell to cell variability of chromatin accessibility as identified from scNOMe-seq datasets. As pointed by reviewer #2, the finding of viability of chromatin accessibility between cells is not surprising nor novel. To gain novel biological insights, additional datasets from substantially more cells may need to be generated and analyzed.

3) Fuller analysis of CpG methylation should be performed to illustrate the benefits of simultaneous measurements of accessible regions and DNA methylation in the same cells.

4) Better documentation of sensitivity and specificity of the method, as pointed out by reviewer #1.

*Reviewer #1:*

In this manuscript by Pott, the author reported a new method to simultaneously profile chromatin accessibility and DNA methylation in single cells. The method is a clever adaptation of an exciting method NOME-seq, which combined de novo GpC methylation and whole genome bisulfite sequencing to determine the nucleosome positions in the genome. The author provided proof of principle by carrying out NOMe-seq in single GM12878 and K562 cells. The author showed that scNOMe-seq could reveal chromatin accessibility as reads with methylated GpCs at DNase hypersensitive sites, and uncover cell to cell variability of the chromatin accessibility. He also showed that the method could reveal footprints of transcription factor binding such as CTCF in individual cells. Finally, he provided some evidence that nucleosome spacing could be inferred from the single cell NOMe-seq data. Overall, the author provided preliminary data to support the feasibility and utilities of the scNOMe-seq method. Important concerns with some technical aspects of the work will need to be addressed before it could be considered as suitable for publication.

1) The author stated that scNOMe-seq allows measurement of chromatin accessibility. This statement is supported by the observation that DNaseI hypersensitive sites mapped from bulk cell population showed increased levels of GpC methylation in single cells. However, the author has yet to provide data on sensitivity and specificity of scNOMe-seq in mapping accessible chromatin regions. From the scNOMe-seq datasets, it is not clear how the accessible regions can be identified. Data analysis method finding such regions is missing. Also lacking is a systematic comparison of the accessible regions identified by scNOMe-seq and DHS mapped from bulk data.

2) The author observed that many DHS show 50% GpC methylation, and concluded that this indicates cell to cell variability of chromatin accessibility. While this is certainly a plausible explanation, the author needs to rule out the possibility of incompleteness of the GpC methylation reaction. In other words, the author needs to document the sensitivity (or false negative rates) of the assay. For example, he could examine the GpC methylation rate at a number of positive control regions where chromatin accessibility should exist in all cells, and use the result to determine the completeness of the GpC methylation reaction.

3) The author stated that "scNOMe-seq detected CTCF DNA binding events from single cells". Again, similar to point #1, this is an overstatement. The author has failed to demonstrate that scNOMe-seq data lead to identification of CTCF binding in individual cells. Aggregate analysis over many CTCF binding sites identified from bulk ChIP-seq data does not go far enough. It is yet not clear how sensitive and specific scNOMe-seq datasets could allow one to map CTCF binding events in single cells.

4) In Discussion, the author stressed advantages of scNOMe-seq over other single cell techniques, but failed to describe some obvious limiting factors, such as restricted scope of analysis (limited to regions with GpC), costs of analysis, number of cells that could be profiled, etc.

*Reviewer #2:*

In this manuscript, Pott presents single-cell NOMe-seq, a genomic technique that allows the simultaneous measurement of DNA methylation and chromatin accessibility. The author applies scNOMe-seq to GM12878 and K562 cell lines. The analyses show that scNOMe-Seq recapitulates results from bulk NOMe-seq, that scNOMe-seq can distinguish between cell types, that scNOMe-seq can reveal TF footprinting in single cells, and that scNOMe-seq can inform about nucleosome phasing in single cells.

One of the main benefits of scNOMe-seq is the ability to distinguish drop-out events from false negatives, thus allowing a truer single-cell representation than existing single-cell chromatin accessibility methods such as ATAC-Seq. However, this comes at the cost of having to sequence much deeper, and this prohibitive cost could limit the adoption of the scNOMe-seq.

My assessment is that the method is a novel advance. However, the analyses presented here do not convincingly show how the scNOMe-seq is of much greater utility than existing methods. In addition, the current manuscript is lacking in new biological insights.

1) Much of the analyses presented in this manuscript (most of Figure 1–Figure 2 and Figure 4, some of Figure 3) are meta-analyses, where either multiple cells are grouped together or similar genomic features are grouped together. While such analysis is important to show that the data are consistent with bulk NOMe-seq, it does not convey the true utility of single-cell NOMe-seq.

2) The author writes that scNOMe-seq "provided direct evidence for the notion that the degree of DNaseI accessibility observed in DNase-seq of bulk samples reflects the frequency with which a particular region is accessible in individual cells." While this statement is true, it is not entirely novel as the same conclusion could be made from bulk NOMe-seq. Since bulk NOMe-seq gives an absolute measure of CpG and GpC on a scale from 0% to 100%, its measurement gives an estimate of the number of cells in the population that have a CpG or a GpC.

3) Perhaps part of the reason that the manuscript does not reveal new biological insights is that too few cells were analyzed, and the coverage per cell is too low. For example, having enough cells would allow the author to address questions of single-cell heterogeneity.

4) The author spends the vast majority of time analyzing scNOMe-seq for chromatin accessibility. But one of the main utilities of this method is the ability to examine single cell DNA methylation. A fuller analysis between the interplay of DNA methylation and chromatin in single cells could add substantial novelty to the manuscript.

*Reviewer #3:*

The author presents a single-cell version of NOMe-seq, a previously described method that provides simultaneous DNA accessibility and DNA methylation measurements. The single-cell version of this method presented here uses FACs sorting of nuclei and Illumina sequencing in a 96-well format, with results presented from libraries derived from 23 nuclei representing two different lymphoid-der cell types. This is a proof-of-principle exercise, and not intended to break any new ground with respect to biological insights. The analyses are sufficiently thorough to demonstrate that scNOMe-seq results are consistent with results using the original bulk version. The data are impressively clean and the efficiency is quite good considering the inherent sparseness of the single cell approach and the deleterious effects of bisulfite treatment of DNA. I have no technical concerns.

The key question in my mind is whether there is likely to be sufficient interest in this tool from readers of *eLife* to qualify as a Tools and Resources article, which must be "especially important" for the field and have "the potential to accelerate discovery". Bulk NOMe-seq is already a single-molecule method, in that each read represents a short snippet of a single genome from a single cell. Single-cell methods for measuring DNA accessibility alone are already relatively mature, in some cases allowing for orders-of-magnitude more single cells to be profiled than using scNOMe-seq, for instance using ATAC-seq with barcoding or microfluidics strategies (Cusanovitch 2015 and Buenrostro 2015 cited). Another method allows for whole-genome bisulfite sequencing of single cells (Farlik 2015 cited). Admittedly these methods do not provide simultaneous DNA accessibility and methylation information, but the inherent sparseness and small scale of the method, and the ambiguity caused by diploidy as to which genome in the cell is being profiled, would seem to limit the range of questions that scNOMe-seq can uniquely address. The author shows that the two different cell types can be distinguished (although it is possible that this a batch effect) but in the realm of sorting cell types from a mixture, or estimating cell-type heterogeneity, scRNA-seq should continue to dominate. scRNA-seq has the advantage over scNOMe-seq of measuring nucleic acids that the cell has amplified by transcription, so that the sparseness of gene-specific products is much less of an inherent limitation than when there are only 2 copies of each DNA duplex in a cell. If there is some real-life situation in which someone would prefer to use this method as opposed to others already available for DNA accessibility mapping and DNA methylation profiling, then I would be more enthusiastic.

---

## [Author Response]

*[…] While all three reviewers recognize the novelty of the method, they all raised concern regarding the utilities of scNOMe-seq, and lack of novel biological insights. To strengthen the work, the author would need to carry out in depth analysis of the datasets, and expand the analyses to additional cells. The essential revisions are comprehensive and may take some time to complete. Please let us know how you plan to respond to the concerns below and provide an estimate of the time it will take to do so.*

I would like to respond to this point first, before responding point-by-point to the reviewers’ comments. Thus far, few methods have been established for measuring single cell chromatin organization in single cells and none of them are widely used yet. These methods are adaptations of commonly used approaches in which read counts constitute the signal (except for bisulfite sequencing). This can be problematic on the level of individual cells where such count-based signals appear binary and very sparse. To complement these assays, it is therefore important to develop additional methods that provide information on chromatin organization using orthogonal approaches. In this context, I believe that scNOMe-seq will be an important addition to the set of single cell genomic methods. However, NOMe-seq had not been used as widely as, for example, ATAC-seq and DNase-seq, and a detailed proof-of-principle study was needed to gauge it suitability for this task.

This study was primarily designed to test whether scNOMe-seq captures observed bulk features in single cells. As such this study is limited to providing evidence that scNOMe-seq data recapitulate observations in single cells that were previously made in bulk samples. This study confirms that scNOMe-seq indeed measures chromatin accessibility and DNA methylation in single cells. The revised manuscript also includes a more detailed analysis of GpC and CpG methylation within and around genes. I believe that the data presented in this manuscript make a strong case for a wider application and encourage other researchers to apply scNOMe-seq in their single cell studies.

*Essential revisions:*

*1) Much of the analyses presented in this manuscript are meta-analyses, where either multiple cells are grouped together or similar genomic features are grouped together. While such analysis is important to show that the data are consistent with bulk NOMe-seq, it does not convey the true utility of single-cell NOMe-seq. To demonstrate the utility of this method over state of art single cell technique, the author needs to carry out additional analysis, such as identifying accessible chromatin regions from scNOMe-seq data, and carrying out systematic comparison to existing methods.*

As discussed above, this study is intended to provide proof-of-principle that scNOMe-seq can measure chromatin accessibility and endogenous methylation simultaneously in a single cell. The data demonstrated that scNOMe-seq qualitatively recapitulates chromatin accessibility and CpG methylation characteristic for a range of genomic features. Like analyses for single cell ATAC-seq, these analyses relied on previously annotated regions. Because the coverage is very sparse it is not possible to call accessible regions within single cells.

*2) Better characterize the cell to cell variability of chromatin accessibility as identified from scNOMe-seq datasets. As pointed by reviewer #2, the finding of viability of chromatin accessibility between cells is not surprising nor novel. To gain novel biological insights, additional datasets from substantially more cells may need to be generated and analyzed.*

I agree with the comment; these results are indeed what is expected and it is therefore an important confirmation of the method’s ability to measure chromatin accessibility. As pointed out by the reviewer, this study does not use enough cells to gain novel biological insights into cell-to-cell variation in chromatin accessibility. This study was designed as a proof-of-principle study, while it would have been desirable to also have novel biological insights this is not feasible with the limited set of cells and beyond the scope of this study.

*3) Fuller analysis of CpG methylation should be performed to illustrate the benefits of simultaneous measurements of accessible regions and DNA methylation in the same cells.*

A fuller analysis of endogenous CpG methylation has now been included. This includes direct comparison between GpC and CpG methylation within DHS and includes the use of CpG methylation to separate individual cells by cell type (i.e. GM12878 and K562).

*4) Better documentation of sensitivity and specificity of the method, as pointed out by reviewer #1.*

The manuscript now includes a more direct assessment of the sensitivity and specificity. Figure 2 and Figure 2—figure supplement 7 and 8 include comparisons between GpC methylation in DHS to randomized DHS. These analyses show that GpC methylation is strongly enriched in DHS (even those with very low scores) compared to random locations. They also show that the correlation between GpC methylation levels and peak scores in individual cells is completely lost when the regions are randomized.

*Reviewer #1:*

*[…] Important concerns with some technical aspects of the work will need to be addressed before it could be considered as suitable for publication.*

*1) The author stated that scNOMe-seq allows measurement of chromatin accessibility. This statement is supported by the observation that DNaseI hypersensitive sites mapped from bulk cell population showed increased levels of GpC methylation in single cells. However, the author has yet to provide data on sensitivity and specificity of scNOMe-seq in mapping accessible chromatin regions. From the scNOMe-seq datasets, it is not clear how the accessible regions can be identified. Data analysis method finding such regions is missing. Also lacking is a systematic comparison of the accessible regions identified by scNOMe-seq and DHS mapped from bulk data.*

(replicated from above)

As discussed above, this study is intended to provide proof-of-principle that scNOMe-seq can measure chromatin accessibility and endogenous methylation simultaneously in a single cell. The data demonstrated that scNOMe-seq qualitatively recapitulates chromatin accessibility and CpG methylation typically associated with genomic features. Like analyses for single cell ATAC- seq, these analyses relied on previously annotated regions. Because the coverage is very sparse it is not possible to call accessible regions within single cells. Higher coverage per cell will overcome this limitation to some degree.

*2) The author observed that many DHS show 50% GpC methylation, and concluded that this indicates cell to cell variability of chromatin accessibility. While this is certainly a plausible explanation, the author needs to rule out the possibility of incompleteness of the GpC methylation reaction. In other words, the author needs to document the sensitivity (or false negative rates) of the assay. For example, he could examine the GpC methylation rate at a number of positive control regions where chromatin accessibility should exist in all cells, and use the result to determine the completeness of the GpC methylation reaction.*

I agree with the reviewer that the absolute GpC methylation level might be affected by the completeness of the methylation reaction. Given this experimental set up it is not possible to resolve this entirely. In relative terms the data provided in Figure 2 and Figure 2—figure supplement 7, Figure 2—figure supplement 11 does indicate that the methylation rate increases with a measure of DNase- seq based chromatin accessibility (peak scores). Higher cell-to-cell variability is associated with regions with lower DHS score and thus not randomly associated with sites that went unmethylated.

*3) The author stated that "scNOMe-seq detected CTCF DNA binding events from single cells". Again, similar to point #1, this is an overstatement. The author has failed to demonstrate that scNOMe-seq data lead to identification of CTCF binding in individual cells. Aggregate analysis over many CTCF binding sites identified from bulk ChIP-seq data does not go far enough. It is yet not clear how sensitive and specific scNOMe-seq datasets could allow one to map CTCF binding events in single cells.*

I agree that the analyses do not identify CTCF binding site de novo. However, the ‘footprint’ analysis following this statement, and the visual confirmation that GpCs within documented CTCF binding regions were unmethylated (while surrounding GpCs were methylated) makes me confident to state that this technique *detected* binding events. Because this evidence is presented after the initial statement I modified that statement to ‘These analyses provided evidence that, in aggregate, scNOMe-seq detected chromatin accessibility characteristic of CTCF binding in single cells.’

*4) In Discussion, the author stressed advantages of scNOMe-seq over other single cell techniques, but failed to describe some obvious limiting factors, such as restricted scope of analysis (limited to regions with GpC), costs of analysis, number of cells that could be profiled, etc.*

A more thorough discussion of these considerations has been included in the paper.

*Reviewer #2:*

*[…] My assessment is that the method is a novel advance. However, the analyses presented here do not convincingly show how the scNOMe-seq is of much greater utility than existing methods. In addition, the current manuscript is lacking in new biological insights.*

*1) Much of the analyses presented in this manuscript (most of Figure 1–Figure 2 and Figure 4, some of Figure 3) are meta-analyses, where either multiple cells are grouped together or similar genomic features are grouped together. While such analysis is important to show that the data are consistent with bulk NOMe-seq, it does not convey the true utility of single-cell NOMe-seq.*

As discussed above, I agree with the reviewer that this is a limitation of the study. However, I argue that establishing the scNOMe-seq assay and demonstrating that this method captures features of chromatin organization observed in bulk through various methods is an important advance that will motivate additional studies.

*2) The author writes that scNOMe-seq "provided direct evidence for the notion that the degree of DNaseI accessibility observed in DNase-seq of bulk samples reflects the frequency with which a particular region is accessible in individual cells." While this statement is true, it is not entirely novel as the same conclusion could be made from bulk NOMe-seq. Since bulk NOMe-seq gives an absolute measure of CpG and GpC on a scale from 0% to 100%, its measurement gives an estimate of the number of cells in the population that have a CpG or a GpC.*

The reviewer is correct in pointing out that these measurements (like those obtained from conventional bisulfite sequencing) are single molecule methods and each fragment therefore represents a measurement from a single cell. However, there are many situations in which a population measurement is not informative (or less informative than data obtained from individual cells). For example, scNOMe-seq will be a useful application to characterize cells in heterogeneous mixtures like brain tissue. While performing bulk NOMe-seq would provide a precise measurement of the average accessibility and DNA methylation at a particular locus, in these samples it would reflect the average measurement of the heterogeneous population.

Using scNOMe-seq it would be possible to first group cells based on their similarity (e.g. use GpC or CpG methylation) and then compare the GpC methylation/chromatin accessibility between groups. This analysis is not possible in bulk using bulk NOMe-seq.

*3) Perhaps part of the reason that the manuscript does not reveal new biological insights is that too few cells were analyzed, and the coverage per cell is too low. For example, having enough cells would allow the author to address questions of single-cell heterogeneity.*

The limited number of cells analyzed, the collection of cells in the G1 phase, and the sparse coverage make it difficult to address questions of cellular heterogeneity using these data. As discussed above this study was designed to provide proof-of-principle that NOMe-seq can be applied to single cells. Addressing the question of single-cell heterogeneity is the express motivation behind the development of this approach, but beyond the scope of the current study.

*4) The author spends the vast majority of time analyzing scNOMe-seq for chromatin accessibility. But one of the main utilities of this method is the ability to examine single cell DNA methylation. A fuller analysis between the interplay of DNA methylation and chromatin in single cells could add substantial novelty to the manuscript.*

The reviewer highlights an important point. The initial manuscript focused almost exclusively on chromatin accessibility. A more detailed analysis of CpG methylation has been included as well as a direct comparison of GpC and CpG methylation at individual DHS.

*Reviewer #3:*

*[…] The key question in my mind is whether there is likely to be sufficient interest in this tool from readers of eLife to qualify as a Tools and Resources article, which must be "especially important" for the field and have "the potential to accelerate discovery". Bulk NOMe-seq is already a single-molecule method, in that each read represents a short snippet of a single genome from a single cell. Single-cell methods for measuring DNA accessibility alone are already relatively mature, in some cases allowing for orders-of-magnitude more single cells to be profiled than using scNOMe-seq, for instance using ATAC-seq with barcoding or microfluidics strategies (Cusanovitch 2015 and Buenrostro 2015 cited). Another method allows for whole-genome bisulfite sequencing of single cells (Farlik 2015 cited). Admittedly these methods do not provide simultaneous DNA accessibility and methylation information, but the inherent sparseness and small scale of the method, and the ambiguity caused by diploidy as to which genome in the cell is being profiled, would seem to limit the range of questions that scNOMe-seq can uniquely address. The author shows that the two different cell types can be distinguished (although it is possible that this a batch effect) but in the realm of sorting cell types from a mixture, or estimating cell-type heterogeneity, scRNA-seq should continue to dominate. scRNA-seq has the advantage over scNOMe-seq of measuring nucleic acids that the cell has amplified by transcription, so that the sparseness of gene-specific products is much less of an inherent limitation than when there are only 2 copies of each DNA duplex in a cell. If there is some real-life situation in which someone would prefer to use this method as opposed to others already available for DNA accessibility mapping and DNA methylation profiling, then I would be more enthusiastic.*

I thank the reviewer for these comments. The reviewer correctly pointed out that this is a proof-of-principle study and the data are not providing biological insights. I believe that scNOMe-seq has the ‘potential to accelerate discovery’ and that presentation of these data will encourage further studies using this method. One interesting application for scNOMe-seq would be the identification and characterization of epigenomic subtypes in cells within complex tissues. One approach could be to first use the endogenous CpG methylation to group cells based on their ‘epigenetic’ make up and then to combine the information of all cells within a given group to obtain rich cell-type specific datasets on chromatin organization and DNA methylation.